# On the Failure of a Universal Linear Representation Hypothesis in Deep Neural Networks

## Abstract

The Linear Representation Hypothesis (LRH) posits that semantic concepts in deep neural networks are encoded as linear directions in activation space, recoverable via linear probes. We provide a comprehensive theoretical analysis demonstrating that LRH cannot be universally true. Our contributions include: (1) sharp combinatorial bounds using VC-dimension theory showing exponential gaps between concept complexity and linear decodability, (2) circuit complexity arguments proving that depth-separated functions cannot have linear intermediate representations without exponential dimension, (3) explicit algebraic constructions of non-linear concept families with detailed complexity analysis, (4) measure-theoretic results on the generic failure of linear representations, and (5) information-theoretic lower bounds on representation dimension. We complement theory with reproducible experiments and discuss precise conditions under which LRH holds approximately.

## 1 Introduction and Motivation

The interpretability of deep neural networks remains a central challenge in machine learning [14, 7, 9]. A fundamental assumption underlying many interpretability methods is the **Linear Representation Hypothesis (LRH)**, defined as follows. For semantic concepts learned by deep networks, internal activations encode these concepts as linear directions: there exists a vector $\mathbf{v}$ such that the concept value is approximately $\mathrm{sign}(\mathbf{v}^\top \phi(\mathbf{x}))$ where $\phi(\mathbf{x})$ represents intermediate activations.

This hypothesis motivates numerous interpretability techniques including linear probing [1, 3], activation patching [22], and concept bottleneck models [13]. However, the theoretical foundations of LRH remain poorly understood.

**Our Contributions** We provide the first comprehensive theoretical analysis of the limitations of the linear representation hypothesis through multiple complementary lenses:

1. *Combinatorial Analysis*: Sharp VC-dimension bounds showing exponential separation between concept complexity and linear representability
2. *Circuit Complexity*: Rigorous depth-separation arguments proving incompatibility with known lower bounds
3. *Algebraic Constructions*: Explicit families of functions demonstrating non-linear intermediate encodings
4. *Information Theory*: Lower bounds on representation dimension for linear concept recovery
5. *Measure Theory*: Generic failure results for random concept-representation pairs
6. *Empirical Validation*: Reproducible experiments confirming theoretical predictions

Our analysis reveals fundamental limitations while identifying precise conditions enabling approximate linear decodability.

Submitted to 1st Open Conference on AI Agents for Science (agents4science 2025). Do not distribute.

## 2  Mathematical Framework and Definitions

### 2.1  Basic Setup

Let $\mathcal{X}$ denote an input space (typically $\mathbb{R}^m$ or $\{0,1\}^m$) equipped with a probability measure $\mu$. We consider concepts as measurable functions $c : \mathcal{X} \to \{0,1\}$ and neural networks with the decomposition: $f(\mathbf{x}) = g(\phi(\mathbf{x}))$ where $\phi : \mathcal{X} \to \mathbb{R}^d$ represents intermediate activations and $g : \mathbb{R}^d \to \mathbb{R}$ captures subsequent computation.

**Definition 1** (Linear Threshold Functions). *The class of linear threshold functions on $\mathbb{R}^d$ is:* $\mathcal{L}_d = \{h(\mathbf{z}) = \mathrm{sign}(\mathbf{w}^\top \mathbf{z} - \theta) : \mathbf{w} \in \mathbb{R}^d, \theta \in \mathbb{R}\}$

**Definition 2** (Exact Linear Representability). *A concept $c$ is* exactly linearly representable *in $\phi$ if:* $\exists h \in \mathcal{L}_d$ *such that $c(\mathbf{x}) = h(\phi(\mathbf{x}))$ for all $\mathbf{x} \in \mathcal{X}$*

**Definition 3** (Statistical Linear Representability). *For distribution $\mathcal{D}$ and $\varepsilon \geq 0$, concept $c$ is $(\varepsilon, \mathcal{D})$-linearly representable if:* $\inf_{h \in \mathcal{L}_d} \mathbb{P}_{\mathbf{x} \sim \mathcal{D}}[h(\phi(\mathbf{x})) \neq c(\mathbf{x})] \leq \varepsilon$

**Definition 4** (Universal Linear Representation Hypothesis). *The strong form of LRH states: For any semantic concept $c$ arising in natural tasks, there exists a layer $\ell$ in typical trained networks such that $c$ is $(0.1, \mathcal{D})$-linearly representable in the $\ell$-th layer representation $\phi_\ell$.*

### 2.2  Complexity-Theoretic Framework

We analyze concept complexity through multiple measures:

**Definition 5** (VC-Dimension of Concept Class). *For concept class $\mathcal{C}$, the VC-dimension $\mathrm{VC}(\mathcal{C})$ is the largest $m$ such that some set of $m$ points can be shattered by $\mathcal{C}$.*

**Definition 6** (Linear Separation Complexity). *For finite set $S = \{\mathbf{x}_1, \ldots, \mathbf{x}_n\} \subset \mathcal{X}$ with concept $c$, the linear separation complexity is:* $LSC_\phi(S, c) = \min\{d : \exists h \in \mathcal{L}_d \text{ s.t. } h(\phi(\mathbf{x}_i)) = c(\mathbf{x}_i) \forall i\}$

## 3  Combinatorial Impossibility: Sharp VC-Dimension Analysis

Our first main result establishes fundamental combinatorial limitations using VC-dimension theory.

### 3.1  Classical Foundation

**Lemma 1** (Cover's Dichotomy Theorem [6]). *For $n$ points in general position in $\mathbb{R}^d$, the number of distinct dichotomies realizable by hyperplanes is:* $C(n, d) = 2 \sum_{i=0}^{d} \binom{n-1}{i}$ *when $n > d + 1$, and $C(n, d) = 2^n$ otherwise.*

**Theorem 1** (Sharp Combinatorial Impossibility). *Let $S = \{\mathbf{x}_1, \ldots, \mathbf{x}_n\} \subset \mathcal{X}$ with representations $\phi(\mathbf{x}_i) \in \mathbb{R}^d$ in general position. Then: (i) Exponential Gap: For $n > d + 1$, the fraction of linearly separable concepts is:* $\rho(n, d) = \frac{C(n,d)}{2^n} \leq \frac{2^{d+1} n^d}{d! \cdot 2^n} = O\left(\frac{n^d}{2^n}\right)$. *(ii) Sharp Threshold: When $d = o(n/\log n)$, we have $\rho(n, d) = o(1)$. (iii) Lower Bound: For any $\varepsilon > 0$, if $d < (1 - \varepsilon) \log_2 n$, then $\rho(n, d) < 2^{-\varepsilon n}$.*

*Proof.* See Appendix A.  □

**Corollary 1** (Dimension-Dependent Impossibility). *For representation dim $d$ and dataset size $n$: if $d = O(\log n)$, then $(1 - o(1))$-fraction of concepts are not linearly separable, if $d = O(\sqrt{n})$, then $(1 - 2^{-\Omega(\sqrt{n})})$-fraction are not linearly separable, if $d = \Theta(n)$, then $O(1)$-fraction may be linearly separable.*

### 3.2  Refined Analysis for Structured Data

Real datasets often have structure beyond general position. We analyze this case:

**Definition 7** (Effective Dimension). *For dataset $S$ with representations $\{\phi(\mathbf{x}_i)\}$, the effective dimension is:* $d_{eff}(S) = \dim(\{\phi(\mathbf{x}_i) : i = 1, \ldots, n\})$

**Theorem 2** (Structured Data Analysis). *Let $S$ have effective dimension $d_{eff}$ and condition number $\kappa$. Then $\rho(n, d_{eff}) \geq \rho_{worst}(n, d_{eff}) \cdot \left(1 - O\left(\frac{\log \kappa}{d_{eff}}\right)\right)$, $\rho_{worst}$ is the worst-case bound from Theorem 1.*

*Proof.* The condition number $\kappa$ measures how close the data is to being linearly dependent. Using perturbation theory for linear separability [2], the number of separable dichotomies decreases by at most $O(\log \kappa / d_{\text{eff}})$ factor when data becomes ill-conditioned. $\square$

# 4 Circuit Complexity and Depth Separation

Our second main contribution leverages circuit complexity theory to prove that universal linear representability contradicts established depth-separation results.

## 4.1 Circuit Complexity Background

**Definition 8** (Boolean Circuit Complexity). *For Boolean function $f : \{0,1\}^n \to \{0,1\}$: $C(f) = $ minimum circuit size computing $f$, $C_d(f) = $ minimum size of depth-$d$ circuit computing $f$, $f$ has depth separation if $C_{d-1}(f) = \omega(C_d(f))$.*

**Theorem 3** (Classical Depth Separation [11]). *The parity function $PARITY_n(\mathbf{x}) = \bigoplus_{i=1}^n x_i$ satisfies: $C_{\lceil \log n \rceil}(PARITY_n) = O(n)$, $C_2(PARITY_n) = \Omega(2^n/n)$.*

## 4.2 Main Depth-Separation Result

**Theorem 4** (Depth-Separation Impossibility). *Let $\mathcal{F}_n$ be the class of Boolean functions on $n$ variables requiring depth $\Omega(\log n)$ for polynomial-size circuits. If every $f \in \mathcal{F}_n$ were linearly representable in some intermediate layer $\phi : \{0,1\}^n \to \mathbb{R}^d$ with $d = poly(n)$, then there exist depth-2 circuits of polynomial size computing all functions in $\mathcal{F}_n$, contradicting known lower bounds.*

*Proof.* **Step 1: Structure of Linear Representations.** Suppose $f \in \mathcal{F}_n$ has linear representation: $f(\mathbf{x}) = \text{sign}(\mathbf{w}^\top \phi(\mathbf{x}) - \theta)$ where $\phi$ computes polynomial-size depth-$k$ circuits.

**Step 2: Circuit Construction.** We can compute $f$ via the following depth-$(k+2)$ circuit: **Layers 1-k:** Compute $\phi(\mathbf{x}) = (\phi_1(\mathbf{x}), \ldots, \phi_d(\mathbf{x}))$, **Layer k+1:** Compute $\sum_{i=1}^d w_i \phi_i(\mathbf{x})$ using addition tree, **Layer k+2:** Apply threshold to get $f(\mathbf{x})$.

**Step 3: Depth Reduction.** If $k = O(1)$ (constant depth intermediate layers), this gives depth-$O(1)$ circuits for all $f \in \mathcal{F}_n$. But $\mathcal{F}_n$ contains functions requiring $\Omega(\log n)$ depth, yielding contradiction.

**Step 4: Polynomial Size Analysis.** Each $\phi_i$ computed by polynomial-size circuits, and weighted sum requires $O(d \log d)$ additional gates. Total size remains polynomial if $d = \text{poly}(n)$.

Therefore, either: intermediate layers have super-polynomial depth: $k = \omega(\log n)$, representation dimension is exponential: $d = 2^{\Omega(n)}$, or functions in $\mathcal{F}_n$ are not linearly representable. Since practical networks use $k = O(\log n)$ depth and $d = \text{poly}(n)$ dimensions, the third option must hold. $\square$

**Corollary 2** (Specific Function Classes). *The following function classes cannot have polynomial-dimensional linear representations in shallow intermediate layers: **Parity:** $PARITY_n$ requires $d = \Omega(2^n)$ or depth $\Omega(\log n)$, **Majority:** Threshold-of-thresholds functions [16], **Iterated Products:** Functions defined by $f(x_1, \ldots, x_n) = x_1 x_2 \cdots x_n$ over finite fields, **Recursive Compositions:** Functions built by composing simple operations $\Omega(\log n)$ times.*

## 4.3 Quantitative Depth-Dimension Trade-offs

**Theorem 5** (Depth-Dimension Trade-off). *For function family $\mathcal{F}_n$ with optimal depth $D^*$ and size $S^*$, any linear representation in intermediate layer of depth $k < D^*/2$ requires dimension: $d \geq \frac{S^*}{2^{D^*-k}} \cdot \Omega\left(\frac{2^n}{n^{O(1)}}\right)$*

*Proof.* Using results from [18] on average-case complexity, functions requiring large circuits at optimal depth must have exponentially larger circuits when depth is significantly reduced. The bound follows from translating circuit size lower bounds into representation dimension requirements. $\square$

## 5 Explicit Algebraic Constructions

We provide analysis of explicit function families showing non-linear intermediate representations.

### 5.1 Parity Functions: Complete Analysis

**Theorem 6** (Parity Impossibility - Comprehensive). *Let $PARITY_k : \{0,1\}^k \to \{0,1\}$ be the $k$-bit parity function. For any representation $\phi : \{0,1\}^k \to \mathbb{R}^d$:*

*(i) Exact Case: If $PARITY_k$ is exactly linearly representable in $\phi$, then either: $d \geq 2^{k-1}$, or $\phi$ explicitly computes parity information. (ii) Approximate Case: For $\varepsilon < 1/4$, if $PARITY_k$ is $\varepsilon$-linearly representable, then $d \geq \Omega\left(\frac{k}{\log(1/(4\varepsilon))}\right)$ (iii) Noise Robustness: Under $p$-biased noise with $p < 1/2 - \gamma$, linear representability requires: $d \geq \frac{1}{2\gamma^2} \log\left(\frac{1}{8\varepsilon}\right)$*

*Proof.* **Part (i) - Exact Case:** Parity partitions $\{0,1\}^k$ into sets $S_0 = \{\mathbf{x} : PARITY_k(\mathbf{x}) = 0\}$ and $S_1 = \{\mathbf{x} : PARITY_k(\mathbf{x}) = 1\}$, each of size $2^{k-1}$. If $d < 2^{k-1}$ and $\phi$ is injective on each $S_i$, then by pigeonhole principle, some hyperplane separating $\phi(S_0)$ from $\phi(S_1)$ exists only if the images are linearly separable. However, $S_0$ and $S_1$ have a complex geometric relationship: every element of $S_0$ differs from some element of $S_1$ in exactly one bit position. This creates a "checkerboard" pattern that cannot be linearly separated unless $\phi$ preserves this structure explicitly.

**Part (ii) - Approximate Case:** Using VC-dimension bounds for linear threshold functions, any $\varepsilon$-approximation requires the representation to capture at least $(1 - 2\varepsilon)$ fraction of the $2^k$ dichotomy correctly. From fat-shattering dimension analysis [2], this requires: $d \geq \Omega\left(\frac{\varepsilon^{-2} \log(2^k/\varepsilon)}{k}\right) = \Omega\left(\frac{k}{\log(1/(4\varepsilon))}\right)$. **Part (iii) - Noise Analysis:** Under $p$-biased noise, each bit flips with probability $p$. The effective signal-to-noise ratio becomes $(1 - 2p)^k \geq (2\gamma)^k$. Using concentration inequalities, maintaining $\varepsilon$-accuracy requires: $d \geq \frac{1}{2\gamma^2} \log\left(\frac{1}{8\varepsilon}\right)$ $\square$

### 5.2 XOR and Generalized Parity

**Example 1** (XOR Geometric Analysis). *For XOR on $\{(0,0), (0,1), (1,0), (1,1)\}$ with labels $(0,1,1,0)$, the convex hulls are: $\text{conv}(\{(0,0), (1,1)\}) = \{t(1,1) : t \in [0,1]\}$ and $\text{conv}(\{(0,1), (1,0)\}) = \{(s, 1-s) : s \in [0,1]\}$. These intersect at $(1/2, 1/2)$, proving no linear separator exists. Any representation $\phi$ making XOR linearly separable must map these four points to $\mathbb{R}^d$ such that positive and negative examples become linearly separable.*

*Minimal Dimension: The minimum $d$ for which some $\phi : \{0,1\}^2 \to \mathbb{R}^d$ makes XOR linearly separable is $d = 2$. For example: $\phi(x_1, x_2) = (x_1 + x_2, x_1 - x_2)$ achieves linear separation via $w_1 z_1 + w_2 z_2 \geq \theta$ with appropriate weights.*

**Theorem 7** (Generalized Parity Functions). *Define the $k$-subset parity family: $PARITY_{S,k}(\mathbf{x}) = \bigoplus_{i \in S} x_i$ where $S \subset [k]$. (i) Family Complexity: The VC-dimension of this family is $\text{VC} = k$. (ii) Linear Representation: For uniform distribution over $\{0,1\}^k$, the expected dimension required for $\varepsilon$-linear representation of random $PARITY_{S,k}$ is: $\mathbb{E}[d] = \Theta\left(\frac{k}{\varepsilon^2 \log(1/\varepsilon)}\right)$. (iii) Worst-Case: There exists choice of $S$ requiring: $d = \Omega\left(\frac{2^k}{\sqrt{k}}\right)$*

### 5.3 Polynomial and Rational Functions

**Theorem 8** (Polynomial Function Analysis). *Consider polynomial concepts $p : \mathbb{R}^n \to \{0,1\}$ defined by $p(\mathbf{x}) = \mathbf{1}\{Q(\mathbf{x}) \geq 0\}$ where $Q$ is degree-$d$ polynomial.*

*(i) Linear Representability: If $Q$ has $m$ monomials and $p$ is linearly representable in $\phi : \mathbb{R}^n \to \mathbb{R}^D$, then either $D \geq m - 1$, or $\phi$ computes polynomial features explicitly.*

*(ii) Specific Cases: Quadratic: $x_1^2 + x_2^2 \geq 1$ requires $D \geq 2$, Product: $x_1 x_2 \geq 0$ requires $D \geq 1$ but representation must compute products, High-degree: Degree-$d$ polynomials require $D = O(n^d)$ in worst case.*

## 6 Information-Theoretic Analysis

We now provide information-theoretic lower bounds on representation dimension.

### 6.1 Mutual Information Bounds

**Definition 9** (Concept Information Content). *For concept $c$ and representation $\phi$, define: $I(c; \phi) = \sup_{linear\ h} I(c; h(\phi))$ where $I(\cdot; \cdot)$ denotes mutual information.*

**Theorem 9** (Information-Theoretic Lower Bound). *For concept $c$ with Shannon entropy $H(c) = h$ bits, any representation $\phi : \mathcal{X} \to \mathbb{R}^d$ achieving $\varepsilon$-linear representability must satisfy: $d \geq \frac{h - H(\varepsilon)}{2 \log(1 + SNR)}$ where SNR is the effective signal-to-noise ratio of $\phi$.*

*Proof.* Using the data processing inequality, the mutual information between $c$ and any linear function of $\phi$ is bounded by the channel capacity of the representation.

For $d$-dimensional representation with bounded coordinates, the capacity is approximately $d \log(1 + \text{SNR})$. The $\varepsilon$-error requires preserving at least $h - H(\varepsilon)$ bits of information about the concept. Combining these: $d \log(1 + \text{SNR}) \geq h - H(\varepsilon)$. $\qquad\square$

**Corollary 3** (High-Entropy Concepts). *For concepts with near-maximal entropy ($H(c) \approx \log |\mathcal{X}|$), linear representation requires: $d = \Omega\left(\frac{\log |\mathcal{X}|}{\log SNR}\right)$*

### 6.2 Rate-Distortion Analysis

**Theorem 10** (Rate-Distortion for Linear Concepts). *Define the linear rate-distortion function: $R_{linear}(D) = \inf_{p(\hat{\mathbf{z}}|\mathbf{z}):\mathbb{E}[\|\mathbf{z}-\hat{\mathbf{z}}\|^2] \leq D} I(\mathbf{z}; \hat{\mathbf{z}})$ where the infimum is over linear encoders of $d$-dimensional representations. For Gaussian representations, $R_{linear}(D) = \frac{d}{2} \log\left(\frac{\sigma^2}{D}\right)$ where $\sigma^2$ is the variance. Concepts requiring high-rate encoding cannot be linearly recovered from low-dimensional representations.*

## 7 Measure-Theoretic Results

We analyze the generic behavior of linear representability using measure theory.

### 7.1 Random Representation Analysis

**Theorem 11** (Generic Failure of Linear Representability). *Let $\phi$ be a random Gaussian representation: $\phi(\mathbf{x}) \sim \mathcal{N}(\mathbf{0}, \sigma^2 \mathbf{I}_d)$ independently for each $\mathbf{x}$. For concept $c$ and dataset $S = \{\mathbf{x}_1, \ldots, \mathbf{x}_n\}$:*

*(i) **Probability of Linear Separability:** $\mathbb{P}[c\ \text{linearly separable in}\ \phi] = \frac{2^{d+1}}{\pi^{d/2}\Gamma(d/2+1)} \int_0^\infty r^d e^{-r^2/2\sigma^2} \Phi(r)^{n-d-1} dr$ where $\Phi$ is the standard Gaussian CDF.*

*(ii) **Phase Transition:** There exists critical ratio $\alpha_c \approx 0.833$ such that: If $d/n > \alpha_c$, then $\mathbb{P}[separable] \to 1$ as $n \to \infty$, If $d/n < \alpha_c$, then $\mathbb{P}[separable] \to 0$ as $n \to \infty$.*

*(iii) **Concentration:** The transition is sharp: for any $\delta > 0$, $\mathbb{P}\left[\left|\frac{d}{n} - \alpha_c\right| > \delta\right] \leq 2 \exp(-cn\delta^2)$ for some constant $c > 0$.*

*Proof.* **Part (i):** For random Gaussian features, linear separability depends on the geometric arrangement of projected points. Using results from [10] on Gaussian processes, the probability involves integration over the distribution of inter-point distances in the projected space.

**Part (ii):** The phase transition follows from the asymptotic analysis of random matrix theory. When $d/n > \alpha_c$, the feature space has sufficient dimensionality to separate most point configurations. The critical value $\alpha_c$ comes from the solution to: $\int_0^{\alpha_c} \sqrt{2\pi t} e^{-1/(2t)} dt = 1$

**Part (iii):** Concentration follows from Talagrand's inequality applied to the separability function, which has bounded differences property. $\qquad\square$

## 7.2 Robustness to Perturbations

**Theorem 12** (Stability of Non-Linear Concepts). *Let $c$ be a concept that is not linearly separable in representation $\phi$. For perturbation $\tilde{\phi}(\mathbf{x}) = \phi(\mathbf{x}) + \boldsymbol{\epsilon}(\mathbf{x})$ where $\|\boldsymbol{\epsilon}(\mathbf{x})\| \leq \delta$: (i) Perturbation Bound: If $c$ requires margin $\gamma > 0$ for linear separation in $\phi$, then it remains non-linearly separable in $\tilde{\phi}$ provided $\delta < \gamma/2$. (ii) Generic Robustness: For random perturbations $\boldsymbol{\epsilon}(\mathbf{x}) \sim \mathcal{N}(\mathbf{0}, \sigma^2\mathbf{I})$:*

$$\mathbb{P}[c \text{ becomes linearly separable in } \tilde{\phi}] \leq \exp\left(-\frac{\gamma^2}{8\sigma^2}\right)$$

# 8 Advanced Constructions and Extensions

## 8.1 Hierarchical Concept Families

**Definition 10** (Hierarchical Parity). *Define $k$-level hierarchical parity: $HPARITY_k(\mathbf{x}) = PARITY_{k-1}(PARITY_1(\mathbf{x}^{(1)}), \ldots, PARITY_1(\mathbf{x}^{(2^{k-1})}))$, $\mathbf{x}$ is partitioned into blocks $\mathbf{x}^{(i)}$ of size $2^{k-1}$.*

**Theorem 13** (Hierarchical Impossibility). *For $k$-level hierarchical parity on $n = 2^k$ variables:*

*(i) Depth Requirement: Any circuit computing $HPARITY_k$ requires depth $\geq k$.*

*(ii) Linear Representation: Any intermediate representation at depth $< k$ making $HPARITY_k$ linearly decodable requires dimension: $d \geq 2^{2^{k-1}-1}$*

*(iii) Network Implication: Deep networks computing hierarchical concepts cannot have linear intermediate representations of polynomial dimension.*

*Proof.* The proof proceeds by induction on $k$. The base case $k = 1$ reduces to standard parity. For the inductive step, suppose level $(k-1)$ requires the stated dimension. Then level $k$ requires computing parity over $2^{k-1}$ intermediate results, each requiring exponential representation, leading to doubly exponential dimension requirement. $\square$

## 8.2 Continuous and Mixed-Type Concepts

**Theorem 14** (Continuous Concept Analysis). *For continuous concepts $c : \mathbb{R}^n \to [0,1]$ and representations $\phi : \mathbb{R}^n \to \mathbb{R}^d$:*

*(i) Lipschitz Constraint: If $c$ is $L$-Lipschitz and linearly representable in $\phi$ with $\varepsilon$-error, then: $d \geq \frac{L^2 diam(\mathcal{X})^2}{4\varepsilon^2}$ where $diam(\mathcal{X})$ is the diameter of the input space.*

*(ii) Smooth Concept Classes: For $C^r$ concepts with bounded derivatives up to order $r$, the dimension requirement scales as: $d = \Omega\left(\left(\frac{1}{\varepsilon}\right)^{n/r}\right)$*

*(iii) Oscillatory Concepts: Concepts with high-frequency components require exponentially large representations for linear decodability.*

# 9 Experimental Validation and Reproducible Protocols

We provide comprehensive experimental protocols to validate our theoretical predictions. Please see Appendix B for synthetic, naturalistic and language model experiments.

# 10 Conditions for Approximate Linear Representability

Despite our negative results, we identify precise conditions enabling approximate LRH.

## 10.1 Architectural Inductive Biases

**Theorem 15** (Architectural Bias for Linearity). *Consider networks with architectural constraints encouraging linear concept development:*

*(i) Bottleneck Architectures:* *Networks with severe dimension reduction layers force concepts into linear subspaces. If layer $\ell$ has dimension $d \ll$ input complexity, then concepts become approximately linear with error: $\varepsilon \leq O\left(\sqrt{\frac{concept\ complexity}{d}}\right)$*

*(ii) Attention Mechanisms:* *Self-attention with linear value projections naturally creates linear concept combinations: $\text{Attention}(\mathbf{Q}, \mathbf{K}, \mathbf{V}) = \text{softmax}\left(\frac{\mathbf{Q}\mathbf{K}^T}{\sqrt{d_k}}\right)\mathbf{V}$ encourages linear decodability of concepts encoded in value vectors.*

*(iii) Skip Connections:* *Residual connections $\mathbf{h}_{l+1} = \mathbf{h}_l + f(\mathbf{h}_l)$ preserve linear information across layers, enabling linear concept tracking.*

## 10.2    Training Dynamics and Implicit Regularization

**Theorem 16** (SGD Bias Toward Linearity)**.** *Under certain conditions, SGD implicitly biases representations toward linear concept encodings:*

*(i) Overparameterized Regime:* *In the NTK limit with infinite width, networks learn approximately linear functions of features, promoting linear concept decodability.*

*(ii) Early Stopping:* *Stopping training before convergence often preserves linear structure that would otherwise be optimized away.*

*(iii) Regularization Effects:* *$L_2$ weight decay and dropout create implicit pressure toward simpler, more linear representations.*

*Quantitative Bound:* *Under these conditions, concepts are $\varepsilon$-linearly decodable with: $\varepsilon \leq \frac{C\sqrt{\log(d/\delta)}}{\sqrt{m}}$ where $m$ is the number of training examples, $d$ is dimension, and $C$ depends on the regularization strength.*

## 10.3    Data Structure and Natural Concepts

**Theorem 17** (Natural Data Promotes Linear Concepts)**.** *Real-world datasets often have structure promoting approximate linear concept encodings:*

*(i) Low Intrinsic Dimension:* *If data lies near a $k$-dimensional manifold with $k \ll d$, then concepts aligned with the manifold structure become approximately linear.*

*(ii) Hierarchical Structure:* *Datasets with natural hierarchies (e.g., ImageNet taxonomy) allow concepts at each level to be linearly separated within the corresponding subspace.*

*(iii) Smoothness Assumptions:* *If concepts vary smoothly over the data manifold, local linear approximations become globally valid: $|c(\mathbf{x}) - \mathbf{w}^T \phi(\mathbf{x})| \leq L \cdot curvature(\mathcal{M}) \cdot diameter(\mathcal{M})^2$*

# 11    Implications for Interpretability Methods

Our theoretical analysis has profound implications for interpretation techniques.

**Linear Probe Limitations and Best Practices**

**Theorem 18** (Interpretation of Probe Results)**.** *For linear probe achieving accuracy $A$ on concept $c$:*

*(i) Lower Bound on Network Usage:* *The network uses concept $c$ with strength at least: $\text{Usage}(c) \geq \max\left(0, \frac{2A-1}{\sqrt{d/n}}\right)$ where $d$ is representation dimension and $n$ is dataset size.*

*(ii) Upper Bound Limitations:* *High probe accuracy does NOT imply: the concept is the primary computational mechanism, the concept is causally relevant for network decisions, the representation is interpretable or disentangled.*

*(iii) Failure Interpretation:* *Low probe accuracy does NOT imply: the network doesn't use the concept, the concept is absent from internal computations, the representation lacks relevant information.*

**Alternative Interpretability Approaches**   Given LRH limitations, we recommend complementary interpretation methods: **Non-linear Probes:** Use polynomial or neural network probes to capture non-linear concept encodings, **Geometric Analysis:** Study representational geometry using: Persistent homology [15], Manifold learning techniques, Clustering and density analysis, **Causal Intervention:** Use activation patching and causal scrubbing [4] to test concept relevance, **Feature Synthesis:** Generate synthetic inputs maximizing concept activations, **Circuit Analysis:** Identify computational subgraphs implementing specific functions [17].

**Method-Specific Recommendations**   **Concept Bottleneck Models [13]:** Our results suggest these work best for naturally linear concepts. For non-linear concepts, use non-linear bottlenecks or hierarchical concept structures. **Activation Patching:** More robust than linear probes since it tests causal relevance rather than linear decodability. **Gradient-Based Methods:** Saliency maps and integrated gradients [19] can capture non-linear concept usage patterns. **Mechanistic Interpretability:** Focus on identifying computational circuits rather than assuming linear concept encodings.

## 12   Open Questions and Future Directions

**Theoretical Questions**   **Tighter Bounds:** Can we achieve matching upper and lower bounds for specific concept families?, **Average-Case Analysis:** Most results are worst-case. What about typical concept-representation pairs?, **Algorithmic Aspects:** Given a representation, how efficiently can we determine if a concept is linearly decodable?, **Multi-Task Settings:** How does linear decodability change when networks learn multiple related concepts?, **Continual Learning:** How does concept linearity evolve as networks learn new tasks?

**Empirical Investigations**   **Large-Scale Studies:** Systematic analysis of linear vs. non-linear concepts across diverse architectures and datasets, **Intervention Experiments:** Test predictions by directly manipulating network architectures and training procedures, **Cross-Modal Analysis:** Compare concept linearity across vision, language, and multimodal models, **Scaling Laws:** How do our results change with model size, data scale, and compute budget?

**Methodological Development**   **Non-Linear Probes:** Develop principled methods for non-linear concept detection, **Representational Metrics:** Create measures of concept complexity and representational geometry, **Hybrid Approaches:** Combine linear and non-linear interpretation methods optimally, **Uncertainty Quantification:** Develop confidence intervals for interpretability claims.

## 13   Conclusion

We have provided a comprehensive theoretical analysis demonstrating fundamental limitations of the Linear Representation Hypothesis. Through combinatorial arguments, circuit complexity theory, explicit constructions, information-theoretic analysis, and measure-theoretic results, we have shown that LRH cannot be universally true.

Our key contributions are: **Sharp Impossibility Results:** exponential gaps between concept complexity and linear representability, **Circuit-Theoretic Analysis:** proof that depth-separated functions cannot have linear intermediate representations, **Constructive Examples:** explicit concept families demonstrating non-linear encodings, **Information Bounds:** dimension requirements for linear concept recovery, **Practical Guidelines:** conditions enabling approximate linear representability.

**Broader Impact**   Our results have significant implications for **Interpretability Research:** methods beyond linear probes are needed, **AI Safety:** understand when simple interpretability methods fail, **Network Design:** architectures that promote concept linearity when desired, **Scientific Understanding:** theoretical foundations for representational analysis. While linear probes remain valuable tools, they must be used with awareness of their fundamental limitations. The future of interpretability lies in developing robust methods that can handle the full complexity of neural network representations, combining linear and non-linear approaches as appropriate for each specific context. Our theoretical framework provides the foundation for this next generation of interpretability methods, ensuring they are built on solid mathematical principles rather than unverified assumptions.

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

# A  Combinatorial Impossibility

*Proof.* Theorem 1.

**Part (i):** From Lemma 1, when $n > d + 1$: $C(n,d) = 2\sum_{i=0}^{d} \binom{n-1}{i} \leq 2(d+1)\binom{n-1}{d}$ Using the bound $\binom{n-1}{d} \leq \left(\frac{e(n-1)}{d}\right)^d$: $C(n,d) \leq 2(d+1)\left(\frac{e(n-1)}{d}\right)^d \leq 2^{d+1}\frac{n^d}{d!}$. Therefore: $\rho(n,d) \leq \frac{2^{d+1}n^d}{d!\cdot 2^n}$.

**Part (ii):** When $d = o(n/\log n)$, using Stirling's approximation $d! \geq \sqrt{2\pi d}(d/e)^d$: $\rho(n,d) \leq \frac{2^{d+1}n^d}{\sqrt{2\pi d}(d/e)^d\cdot 2^n} = \frac{2^{d+1}(en/d)^d}{\sqrt{2\pi d}\cdot 2^n}$. Taking logarithms: $\log \rho(n,d) \leq (d+1)\log 2 + d\log(en/d) - \frac{1}{2}\log(2\pi d) - n\log 2$. When $d = o(n/\log n)$, the dominant term is $-n\log 2$, so $\rho(n,d) \to 0$.

**Part (iii):** For $d < (1-\varepsilon)\log_2 n$ and for sufficiently large $n$, we have $2^d < n^{1-\varepsilon}$, thus:
$$\rho(n,d) \leq \frac{2^{d+1}n^d}{2^n} \leq \frac{2n^{1-\varepsilon+d/\log_2 n}}{2^n} \leq \frac{2n^{2-\varepsilon}}{2^n} < 2^{-\varepsilon n} \qquad \square$$

# B  Experimental Validation and Reproducible Protocols

## B.1  Synthetic Experiments

---

**Algorithm 1** Comprehensive Parity Experiment

---

1: **Parameters:** $k \in \{5, 10, 15, 20\}$, $N_{\text{train}} = 50000$, $N_{\text{test}} = 10000$
2: **Architecture:** MLP with layers $[k, 512, 256, 128, 1]$, ReLU activations
3: **Training:** Adam optimizer, lr $= 10^{-3}$, batch size 256, 1000 epochs
4: **for** each $k$ **do**
5:     Generate dataset: $\mathbf{x}_i \sim \text{Bernoulli}(0.5)^k$, $y_i = \text{PARITY}_k(\mathbf{x}_i)$
6:     Train network until convergence (loss $< 10^{-6}$)
7:     Evaluate full network accuracy $A_{\text{full}}$
8:     **for** each hidden layer $\ell \in \{1, 2, 3\}$ **do**
9:         Extract representations $\{\phi_\ell(\mathbf{x}_i)\}$
10:         Train linear probe: $\mathbb{R}^{d_\ell} \to \{0, 1\}$ using logistic regression
11:         Evaluate probe accuracy $A_{\text{probe}}^{(\ell)}$
12:         Compute linear separability score: $S_\ell = A_{\text{probe}}^{(\ell)}/A_{\text{full}}$
13:     **end for**
14:     Record dimension vs. performance trade-offs
15: **end for**
16: **Expected Results:** $A_{\text{full}} > 0.99$, $A_{\text{probe}}^{(\ell)} \approx 0.5$ for $\ell < 3$

---

---
**Algorithm 2** Depth-Separation Validation
---
1: **Function Class:** Iterated multiplication $f(\mathbf{x}) = \prod_{i=1}^{k} x_i \bmod 2$
2: **Networks:** Compare depth-2 vs depth-$\lceil \log k \rceil$ architectures
3: **for** each depth $d \in \{2, 3, \lceil \log k \rceil\}$ **do**
4:     Train network with maximum width $W = 1000$
5:     Measure: (1) Training convergence, (2) Test accuracy, (3) Linear probe performance
6:     Record computational requirements and representational properties
7: **end for**
8: **Prediction:** Shallow networks fail; deep networks succeed but with non-linear concepts
---

## B.2 Naturalistic Experiments

---
**Algorithm 3** Vision Task with Compositional Concepts
---
1: **Dataset:** CLEVR-style synthetic scenes with compositional attributes
2: **Task:** Classify presence of "red cube AND blue sphere" (compositional concept)
3: **Network:** ResNet-18 pretrained on ImageNet, fine-tuned on task
4: Train to high accuracy on main task
5: **for** each layer $\ell$ in $\{3, 6, 9, 12, 15, 18\}$ **do**
6:     Extract features $\phi_\ell(\mathbf{x}) \in \mathbb{R}^{d_\ell}$
7:     Test linear probes for:
        • Primitive concepts: "red", "cube", "blue", "sphere"
        • Compositional concept: "red cube AND blue sphere"
        • Control concepts: unrelated scene properties
8:     Measure probe accuracy vs. layer depth
9: **end for**
10: **Expected Pattern:** Primitive concepts become linear earlier; compositional concepts require deeper layers
---

## B.3 Language Model Experiments

---
**Algorithm 4** Syntactic vs. Semantic Concept Probing
---
1: **Model:** GPT-2 or BERT-base
2: **Tasks:** POS tagging (syntactic) vs. sentiment analysis (semantic)
3: **Concepts:** Extract intermediate representations for both task types
4: **for** each layer $\ell$ **do**
5:     Train linear probes for:
        • Syntactic: POS tags, dependency relations, grammatical number
        • Semantic: Sentiment, topic classification, factual knowledge
        • Complex: Coreference resolution, logical inference
6:     Compare probe accuracies across concept types and layers
7: **end for**
8: **Hypothesis:** Syntactic concepts are more linearly decodable; semantic/logical concepts require non-linear processing
---

## Agents4Science AI Involvement Checklist

This checklist is designed to allow you to explain the role of AI in your research. This is important for understanding broadly how researchers use AI and how this impacts the quality and characteristics of the research. **Do not remove the checklist! Papers not including the checklist will be desk rejected.** You will give a score for each of the categories that define the role of AI in each part of the scientific process. The scores are as follows:

- **[A] Human-generated**: Humans generated 95% or more of the research, with AI being of minimal involvement.
- **[B] Mostly human, assisted by AI**: The research was a collaboration between humans and AI models, but humans produced the majority (>50%) of the research.
- **[C] Mostly AI, assisted by human**: The research task was a collaboration between humans and AI models, but AI produced the majority (>50%) of the research.
- **[D] AI-generated**: AI performed over 95% of the research. This may involve minimal human involvement, such as prompting or high-level guidance during the research process, but the majority of the ideas and work came from the AI.

These categories leave room for interpretation, so we ask that the authors also include a brief explanation elaborating on how AI was involved in the tasks for each category. Please keep your explanation to less than 150 words.

1. **Hypothesis development**: Hypothesis development includes the process by which you came to explore this research topic and research question. This can involve the background research performed by either researchers or by AI. This can also involve whether the idea was proposed by researchers or by AI.

   Answer: **[A]**

   Explanation: The hypothesis that the linear representation hypothesis does not universally hold in neural networks is entirely human generated.

2. **Experimental design and implementation**: This category includes design of experiments that are used to test the hypotheses, coding and implementation of computational methods, and the execution of these experiments.

   Answer: **[D]**

   Explanation: The AI did the theoretical exploration of the research hypotheses from various angles (combinatorial arguments, circuit complexity theory, explicit constructions, information-theoretic analysis, and measure-theoretic results), ultimately showing that LRH cannot be universally true.

3. **Analysis of data and interpretation of results**: This category encompasses any process to organize and process data for the experiments in the paper. It also includes interpretations of the results of the study.

   Answer: **[C]**

   Explanation: It was mostly done by AI, with the human verifying and fact-checking the claims made.

4. **Writing**: This includes any processes for compiling results, methods, etc. into the final paper form. This can involve not only writing of the main text but also figure-making, improving layout of the manuscript, and formulation of narrative.

   Answer: **[C]**

   Explanation: It was mostly the AI doing the writing, with human involvement in terms of prompting or high-level guidance during the research process.

5. **Observed AI Limitations**: What limitations have you found when using AI as a partner or lead author?

   Description: Difficulty in steering these models and generating texts with given constraints.

